# Understanding the impact of mobility on *Plasmodium* spp. carriage in an Amazon cross-border area with low transmission rate

Hélène Tréhard[1], Lise Musset[2], Yassamine Lazrek[2], Felix Djossou[3], Loïc Epelboin[3,4], Emmanuel Roux[5,6], Jordi Landier[1], Jean Gaudart[7]*, Emilie Mosnier[1,8,9]

**1** Aix Marseille Institute of Public Health ISSPAM, UMR1252 SESSTIM, Aix-Marseille University, Inserm, IRD, Marseille, France, **2** Laboratoire de parasitologie, World Health Organization Collaborating Center for Surveillance of Antimalarial Drug Resistance, Centre Nationale de Référence du Paludisme, Institut Pasteur de la Guyane, Cayenne, French Guiana, **3** Unité des Maladies Infectieuses et Tropicales, Centre Hospitalier de Cayenne, French Guiana, **4** Centre d'Investigation Clinique Antilles Guyane CIC Inserm 1424, Centre Hospitalier de Cayenne, French Guiana, **5** French National Research Institute for Sustainable Development (IRD), ESPACE-DEV, University of Montpellier, University of French West Indies, University of French Guiana, University of La Reunion, Montpellier, France, **6** French National Research Institute for Sustainable Development (IRD), Sentinela International Joint Laboratory, University of Brasilia (UnB), Oswaldo Cruz Foundation (Fiocruz), Montpellier, France, **7** Aix Marseille University, INSERM, IRD, ISSPAM, SESSTIM, UMR1252, APHM, Hop Timone, BioSTIC, Biostatistic & ICT, Marseille, France, **8** Grant Management Office, University of Health Sciences, Phnom Penh, Cambodia, **9** French Agency for Research on AIDS, Viral Hepatitis and Emerging Infectious Diseases (ANRS-MIE), Phnom Penh, Cambodia

\* jean.gaudart@univ-amu.fr

**Data Availability Statement:** The datasets generated and analysed during the current study are not publicly available as specific French

## Abstract

Despite the large reduction in malaria incidence in the last decade, the last kilometre to elimination is often the hardest, especially in international border areas. This study investigated the impact of mobility on *Plasmodium* spp. carriage in people living in a cross-border area in Amazonia with a low malaria transmission rate. We implemented a longitudinal ancillary study in the French Guiana town of St. Georges de l'Oyapock, which is located on the border with Brazil. It was based on data from two transversal surveys performed in October 2017 and October 2018. Data were collected on peri-domestic mobility for food-producing activities, and longer-distance mobility in high-risk areas. Participants were screened for *Plasmodium* spp. carriage using PCR tests, and treated if positive. Vector density around a participant's home was estimated using a previously published model based on remote sensing and meteorological data. The association between *Plasmodium* spp. carriage and mobility was analysed using a generalized additive mixed model. A total of 1,192 inhabitants, aged between 0 and 92 years old, were included. Median age was 18 years in 2017 (IQR [8;35]). *Plasmodium* spp. prevalence in the study population was 7% in 2017 (n = 89) and 3% in 2018 (n = 35). *Plasmodium* spp. carriage was independently associated with i) travel to the adjoining Oiapoque Indigenous Territories in Brazil (OR = 1.76, p = 0.023), ii) the estimated vector density around a participant's home (High *versus* Low risk OR = 4.11, p<0.001), iii) slash-and-burn farming (OR = 1.96, p = 0.013), and iv) age (p = 0.032). Specific surveillance systems and interventions which take into account different types of mobility are needed in cross-border

National Data Protection Commission ("Commission Nationale de l'Informatique et des Libertés" or CNIL) authorization is required for the transfer of personal data, in order to protect participant privacy. The datasets may be obtained from the corresponding author upon reasonable request only with specific authorization from the CNIL (for CNIL authorization contact dpo@univ-amu.fr).

**Funding:** The PALUSTOP study was funded by European Funds for Regional Development, Synergie funding contract N° GY0012082 (to LM as principal investigator and EM as project staff), and by the French Guiana Regional Health Agency (to LM as principal investigator). The funders had no role in study design, data collection and analysis, decision to publish, or preparation of the manuscript.

**Competing interests:** The authors have declared that no competing interests exist.

areas to achieve and maintain malaria elimination (e.g., reactive case detection and treatment in the places visited).

## Introduction

Malaria is a parasitosis caused by *Plasmodium* spp. which is transmitted by mosquitos. In 2021, it was responsible for 247 million malaria cases and 619,000 deaths [1]. Between 2000 and 2019, malaria incidence significantly decreased in certain countries thanks to improved access to care, the development of new drugs and diagnostic tools, and the implementation of more rigorous surveillance and response systems. Today, the disease has been eliminated in 15 countries. Elsewhere, other countries have already reached the 'very low transmission' phase (defined as annual parasite incidence of < 10000 cases per 100000 population and a *Plasmodium falciparum* rate <1%), which is the last step before elimination. However, despite all the interventions implemented, these countries struggle to reach elimination [1–4]. In certain countries, this is partly because of the relatively high incidence of malaria in international border areas, where daily cross-border mobility is a part of life [5,6]. The reasons for cross-border mobility include daily and seasonal work, family visits, purchasing goods, seeking healthcare, schooling, and immigration [5–7]. Accordingly, interventions to reach malaria elimination in these areas must be specifically tailored to these mobile populations [3,8].

French Guiana is a French overseas territory with low malaria transmission (incidence of 55 per 100000 inhabitants and *P. falciparum* rate of 3% in 2020) [9]. Located in the northeast of South America and bordered by Suriname and Brazil, it is populated by several ethnic groups including native indigenous persons (Amerindians), descendants of ancient population movements, as well as recent economic migrants from neighbouring countries [10,11]. The majority of people in French Guiana live on the coast, and dense forest covers 96% of the territory.

*Plasmodium vivax* accounts for 97% of malaria cases in French Guiana. Cases are mainly located in specific areas in the forest and on the borders with Brazil and Suriname along the Oyapock and Maroni rivers, respectively [9,12,13]. The persistence of malaria here is suspected to be linked to cross-border mobility and to illegal gold mining in the French Guiana forest. The latter activity mostly attracts Brazilian citizens to the forest, and consequently to nearby villages along the countries' borders [12,14–16]. In addition to the regional malaria control plan, to date, two pilot interventions have been conducted in French Guiana. The first, called MALAKIT, was started in 2018; it involves the distribution of self-test and treatment kits to undocumented gold miners in Suriname and Brazil before they enter French Guiana [17]. The second intervention, called PALUSTOP, was a before-after study involving two mass testing and treatment (MTAT) campaigns in 2017 and 2018 for the population living in the French Guiana town of Saint Georges de l'Oyapock (SGO), which lies on the border with Brazil [18]. Besides these two interventions, various collaborations have been developed with the Brazilian government to implement cross-border epidemiological surveillance [19] and cross-border public health programmes [20].

The present study focused on the inhabitants of SGO. The highest number of malaria cases occurs there during the dry season (July-December), specifically between October and December [14]. The main vector is *Anopheles darlingi* [21], whose adaptable behaviour to control measures in terms of egg-laying and biting contributes to the persistence of malaria in the area [22,23]. It is strongly suspected that this vector is also present on the Brazilian side of the border [24,25]. Moreover, the same epidemiological pattern is observed in the Oiapoque

municipality, the main Brazilian city across the river [16,19]. The circulation of *P. vivax* has been shown on both sides of the border [26]. In terms of disease control, although both countries mostly implement the same general strategies (i.e., insecticide-impregnated mosquito bed nets, indoor residual spraying, and free access to testing and treatment), unlike French Guiana, drugs targeting *P. vivax* hypnozoites are distributed without prior G6PD deficiency testing in Brazil. After a large decrease in the incidence of malaria between 2007 and 2016 in SGO, epidemics occurred in 2017 and 2018. Local cross-border mobility of inhabitants is strongly suspected of being involved in the area's malaria epidemiology [14].

The objective of this study was to better understand the impact of mobility of people living in SGO on *Plasmodium* spp. carriage, with a view to improving public health measures for malaria elimination in a cross-border context in Amazonia.

## Materials and methods

### Setting

Located in the French Guiana forest on the Oyapock river, SGO is the main town on the border with Brazil (Fig 1). It has a population of 4245 inhabitants (2019 census) [27] and is divided into 15 neighbourhoods. Twelve of these are grouped around the airport and surrounded by the river, slash-and-burn farming fields and the forest (hereafter, 'peri-domestic area'). They constitute the neighbourhoods in the centre of town and the periphery. The remaining three neighbourhoods are isolated and are located in the forest along the river. The neighbourhoods are very diverse in terms of their populations (Brazilian and French citizens from different ethnic groups such as Amerindians, Creoles, French citizens from mainland France, etc.) and housing (permanent houses (i.e., brick-built or other), houses on stilts partially built on the river, and informal housing).

There are very strong links (transport, trade, business, etc.) between SGO and Oiapoque, the main town on the Brazilian side of the border. Oiapoque is much bigger than SGO, with a population of approximately 20 000 inhabitants. The travel time between the two towns is 15 minutes by boat and the same by road. Although both towns are connected to the rest of their respective country by road, they are still far from the nearest cities. It takes at least 2.5 hours by

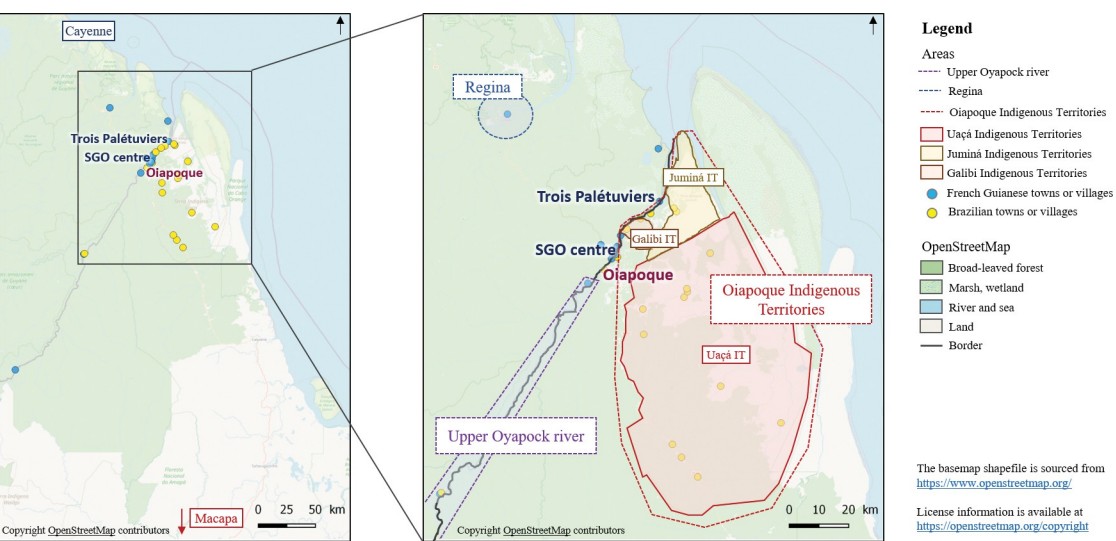

**Fig 1. Oyapock river basin in terms of inhabitation.**

car to reach Cayenne (in French Guiana) and 10 hours by car to reach Macapá (in Brazil) [16] (Fig 1). Apart from SGO and Oiapoque, the Oyapock river basin consists of remote villages scattered along both sides of the river, one village on the way to Cayenne (Regina) and Oiapoque Indigenous Territories (OIT) located along the lower Oyapock river on the Brazilian side, which extend 100 km to the southeast (Fig 1). The latter are historically Amerindian and comprise 57 villages populated by approximately 8000 inhabitants from the same ethnic groups as those in SGO [28–30].

Villages and towns along the border are linked to illegal gold mining in French Guiana. Some of these places serve as supply and stopover points for miners on their way into the forest, particularly in Oiapoque and the upper Oyapock river [31,32]. Slash-and-burn farming, fishing and hunting are the other main activities in the area. Remote villages have local access to primary care and to primary education. Secondary schools and a general hospital are located in both SGO and Oiapoque. Referral hospitals are located in Cayenne and Macapá [33]. Oiapoque is the main town for shopping and nightlife for all the inhabitants of the basin.

Daily life, irrespective of nationality, involves a great deal of cross-border mobility because of the relative isolation of the area and for historical reasons. Amerindians lived in this area long before the current national boundaries between France and Brazil were created [28]. Cross-border mobility happens for the following reasons [34]:

- Daily commuting, especially in the morning and evening, for school, work, and shopping

- Monthly trips for administrative purposes

- Visits to family during the weekend, holidays, and for specific religious festivals.

- Long-term local migration to either side of the border.

- Stopover point in seasonal (i.e., for gold miners) or long-term economic migration by persons form Brazil to cities other than SGO in French Guiana

### Study design

We designed a longitudinal ancillary study based on the before-after study PALUSTOP (see above), which was implemented from September to December 2017 (first survey) and from September to December 2018 (second survey) among a cohort of 1567 people to assess the effectiveness of an MTAT intervention in SGO [18]. Only 12 of the 15 neighbourhoods in SGO were selected for PALUSTOP as malaria prevalence was higher there. These included 11 neighbourhoods in the city centre and periphery, and 1 remote neighbourhood called Trois Palétuviers (Fig 2). All children and adults from these 12 neighbourhoods were invited to participate. Those who agreed were tested for malaria using both a Rapid Diagnostic Test (RDT) and a Polymerase Chain reaction (PCR) test. They also responded to the two survey questionnaires (i.e., 2017 and 2018). Latitude and longitude coordinates of participants' homes were collected. Blood samples were collected by nurses and questionnaires were administered by trained community health workers (CHW) to limit any potential misunderstanding related to cultural and language barriers. In each neighbourhood, the CHW first went door to door, and then brought all those who agreed to participate into one communal location in the neighbourhood, which was specifically set up for PALUSTOP. All persons who then tested positive for either the RDT or PCR received treatment following regional guidelines, whether symptomatic or not [35,36].

Of the town's 4245 inhabitants, 2727 were living in the 12 included neighbourhoods (data from local health centre). A total of 1567 participants were included in the first survey in 2017 (57.42% participation rate).

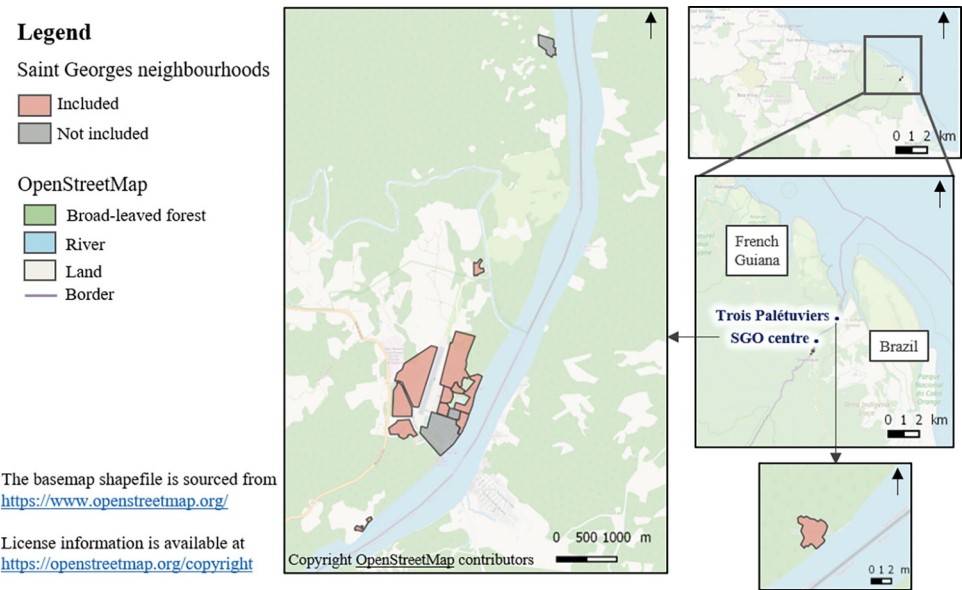

**Fig 2. Study population distribution in the 12 different neighbourhoods of SGO which were included.**

For our longitudinal ancillary study, we focused only on the participants with complete PCR results in both 2017 and 2018 (n = 1192).

## Variables and data sources

***Plasmodium* spp. Carriage.**   The outcome of our ancillary study was *Plasmodium* spp. carriage, defined as a positive PCR for *Plasmodium* spp.. Malaria PCR detection was performed using a real-time method derived from Shokoples et al. [37]. Detection and identification of four *Plasmodium* species (*P. falciparum*, *P. vivax*, *P. ovale*, and *P. malariae*) were performed using a Taqman® (Vilnius, Lithuania) probe strategy, with a sensitivity of 1 parasite/μL for all the species except *P. vivax* at 0.25 parasite/μL.

**Risk factors associated with *Plasmodium* spp. Carriage.**   The potential risk factors associated with *Plasmodium* carriage were divided between factors around the participant's home, and those linked to mobility outside this area, such as travel to high-risk areas, trips to gold mining sites, and outdoor activities in peri-domestic areas (i.e., forest, fields, and tributaries of the river Oyapock around the town/ neighbourhood) for subsistence, work, or recreational activities (slash-and-burn farming, hunting, fishing).

Only one potential risk factor was assessed regarding the area around a participant's home. This was vector density, categorized as low, medium or high; it was assessed by superimposing the geographical coordinates of participants' homes on a vector density map. This map had previously been developed using variables derived from satellite imagery and meteorological observations produced by Adde et al. (2016) [21,22] (Fig 3). For the remote neighbourhood of Trois Palétuviers (Fig 1), all the homes were considered to have high vector density based on a previous entomological study [14].

Risk factors linked to mobility outside the area around one's home were divided into outdoor activities, trips to gold mining sites and travel to high-risk areas. With regard to outdoor activities in peri-domestic areas, people were considered to be at risk if they declared fishing, slash-and-burn farming or hunting often (i.e., more than three times a week) or sometimes.

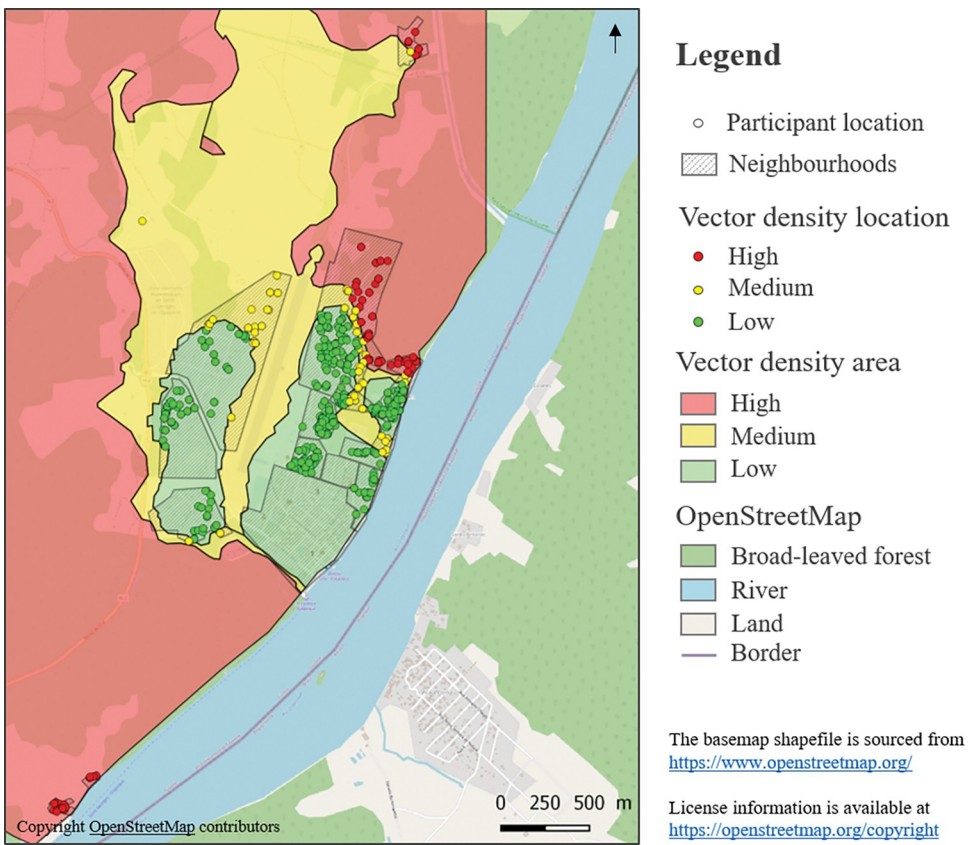

**Fig 3. Estimated vector density based on 'Dynamical Mapping of *Anopheles darlingi* Densities in a Residual Malaria Transmission Area of French Guiana by Using Remote Sensing and Meteorological Data'.** (Adde et al., 2016) [22].

Gold mining is linked to *Plasmodium* spp. carriage in French Guiana [15,31]. Participants who reported visiting a gold mining site at least once were considered to be at risk.

The possible high-risk areas where SGO inhabitants generally travel were split into four areas: the upper Oyapock river (French Guiana and Brazil), Oiapoque (Brazil), Regina (French Guiana) and the OIT (Brazil) (Fig 1). The data regarding the three first areas were collected only during the first survey while the data regarding the last area were collected both in 2017 and 2018. For each area, we used a binary variable to determine the potential risk of *Plasmodium* spp. carriage as follows: at least one visit (yes/no).

**Confounding factors.** The age, sex and ethnic group of participants were taken into account in the analysis, as all three variables have been shown to be potentially linked to *Plasmodium* spp. carriage in the area [16]. Ethnic groups were divided into Amerindians (i.e., speak at least one Amerindian language as their mother tong) and others (i.e., no Amerindian language spoken).

## Statistical analysis

A generalized additive mixed model (GAMM) with logistic regression was built to assess the effect of the different risk factors on *Plasmodium* spp. carriage. With regard to travel in high-risk areas, only the OIT were included as they were the only destination significantly associated with *Plasmodium* spp. carriage. A nested random effect on participant, family and

neighbourhood was added to take into account the effect of possible unassessed common behaviours in the same family and/or neighbourhood. A temporal autocorrelation was applied to consider the repeated measurement in participants. A Gaussian process using a power exponential covariance function was applied to the geographical coordinates of participants' homes to take into account the potential effect of the proximity of homes in the spread of the disease [38]. A spline was used to interpolate the quantitative data in the model (age).

The statistical analyses were performed using R software version 4.1.2. (Copyright 2022, R Foundation for Statistical Computing). The maps in the figures were created using QGIS, version 2.18.28 (Open Source Geospatial Foundation Project, Beaverton, OR, USA). Figures and images were formatted and processed using Microsoft Paint.

### Ethics statements

The PALUSTOP study was approved by the Comité de Protection des Personnes du Sud-Ouest et Outre-Mer 4 N° AM-36/1/CPP15-024. Prior to the study, written informed consent was obtained from all participants. The database of the PALUSTOP prospective study was anonymized and declared to the French regulatory commission (Commission Nationale Informatique et Libertés, CNIL, n°917186).

## Results

### Population characteristics

**Sociodemographic data.**   Of the total 1567 participants included during the first survey of PALUSTOP (i.e., in 2017), only 1277 were assessed during the second survey (i.e., in 2018). Of the 290 participants lost to follow-up, 56% had moved to another area in the intervening period and 7% refused to participate; the remaining 37% could not be reached (24% not found and 13% absent). Only 1192 participants who participated in both surveys had available PCR results for both 2017 and 2018 and fully completed both survey questionnaires (Fig 4). They constituted our study population for the present ancillary study.

Median age of the study population was 18 years in 2017 (IQR [8;35]; range 0 to 92 years), and 55% were female (Table 1). With regard to the location of homes, 14% lived in Trois Palétuviers, while 86% lived in neighbourhoods in the town centre and periphery (Table 1).

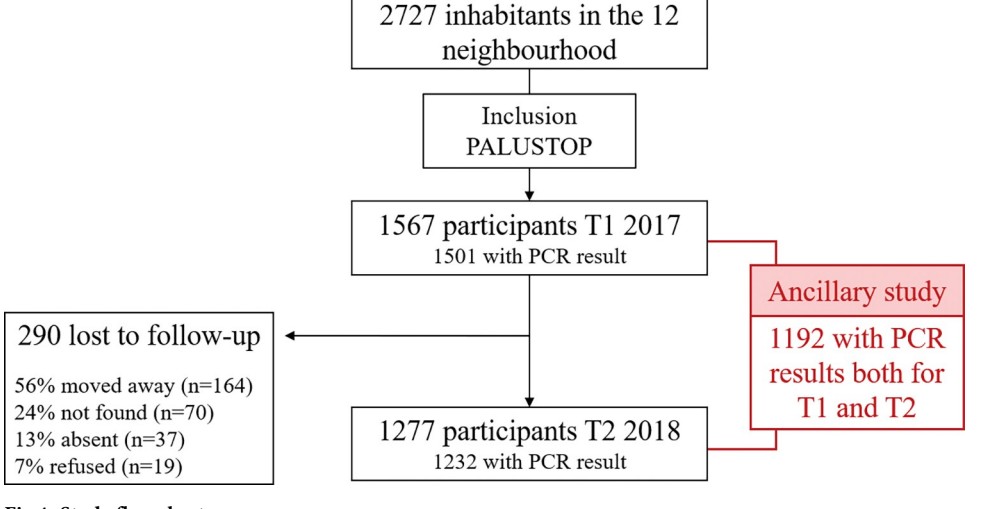

**Fig 4. Study flow chart.**

**Table 1. Sociodemographic characteristics of the study population.**

| Characteristics | Total number of persons (%) |
|---|---|
| **Total** | 1192 (100%) |
| **Male** | 541 (45%) |
| **Age group (years)** | |
| 0–17 | 589 (49%) |
| 18–25 | 137 (12%) |
| 26–55 | 383 (32%) |
| >55 | 83 (7%) |
| **Nationality** | |
| French | 671 (56%) |
| Brazilian | 514 (43%) |
| Other | 6 (0.5%) |
| **Ethnic group** | |
| Amerindian | 812 (68%) |
| Other | 380 (32%) |
| **Neighbourhood** | |
| Trois Palétuviers | 170 (14%) |
| Other included neighbourhoods in Saint Georges de l'Oyapock | 1022 (86%) |

The main occupations of adult participants were homemaker (40%), informal work (19%) and outdoor activities (24%) (slash-and-burn farming (14%), fishing/sailing (5%) or hunting (5%)). In terms of nationality, 56% were French (n = 671) and 43% Brazilian (n = 514), with a significant variation between age groups. The Brazilian participants had lived a median of 11.5 years in SGO (IQR [5;19.75]). In terms of ethnicity, 68% of the participants were Amerindian (n = 812), with a significant variation between neighbourhoods (Tables 1 and S1).

**Risk of exposure description.** In terms of the potential risk of *Plasmodium* spp. carriage related to the area around one's home, 28% (n = 328) participants lived in a house located in a high estimated vector density zone, 14% (n = 161) a medium vector density zone, and 59% a low vector density zone (n = 703).

In terms of the overall risk of *Plasmodium* spp. carriage related to mobility outside the area of one's home, a sizeable proportion of the population (adults and children) practiced outdoor activities, whether for work, for subsistence, or for recreation (Table 2). More specifically, 50%, 30% and 16% practiced slash-and-burn farming, fishing and hunting, respectively. Of

**Table 2. Risk factors linked to mobility outside the area around one's home for *P. vivax* carriage among the whole study population, among children (under 18 years old) and among adults (18 years old and above.**

| | Total population<br>Number (%) | Children (<18 years old)<br>Number (%) | Adults<br>Number (%) |
|---|---|---|---|
| **Total** | 1192 (100%) | 579 (100%) | 603 (100%) |
| **Slash-and-burn farming** | 592 (50%) | 271 (47%) | 320 (53%) |
| **Hunting** | 192 (16%) | 40 (7%) | 152 (25%) |
| **Fishing** | 358 (30%) | 126 (22%) | 232 (39%) |
| **Trips to gold-mining sites** | 40 (3%) | 9 (2%) | 31 (5%) |
| **Travel to Oiapoque Indigenous Territories** | 207 (17%) | 76 (13%) | 130 (22%) |
| **Travel to Oiapoque** | 88 (7%) | 34 (6%) | 54 (9%) |
| **Travel to Upper Oyapock river** | 20 (2%) | 7 (1%) | 13 (2%) |
| **Travel to Regina** | 18 (2%) | 7 (1%) | 11 (2%) |

these, 14%, 35% and 58%, respectively, practiced the activity at night from time to time. The type of outdoor activity practiced differed greatly between the neighbourhoods. Specifically, 28% of the participants living in Trois Palétuviers declared hunting *versus* only 14% in the town centre and periphery neighbourhoods. Moreover, 43% of those living in neighbourhoods along the river (Fig 1) declared fishing *versus* 22% in the other neighbourhoods. The type of outdoor activity was also strongly linked to ethnic group, with more than 61% of Amerindians declaring slash-and-burn farming *versus* 26% of participants from the 'other ethnicity' group.

With regard to other risk factors for mobility outside the area of one's home, 3% of the study population had visited gold mining sites, and 8% had a household member who had visited gold mining sites (Table 2). In terms of travel to high-risk areas, 17% went to the OIT in 2017 and/or 2018 (n = 207), with large variations between the neighbourhoods: from 43% in Trois Palétuviers to 13% in the town centre and periphery neighbourhoods (S1 Table).

### *Plasmodium* spp. carriage prevalence in the study population

Among the 1192 participants in our study population, 89 (7%) had a positive PCR in 2017 (80 for *P. vivax* and 9 for *P. falciparum*), and 35 (3%) in 2018 (30 and 5, respectively); 10 had a positive PCR in both surveys (all 10 were *P. vivax*) (Fig 5).

The factors associated with *Plasmodium* spp. carriage in both years in the bivariate analysis were the neighbourhood of residence (with a higher percentage of cases in Trois Palétuviers), high vector density around the participant's home, travelling to the OIT, slash-and-burn farming, and finally, fishing. Older age was associated with *Plasmodium* spp. carriage only in 2018, while hunting and visiting gold mining sites were only associated with it in 2017 (S2 Table). Taking into account all the risk factors for *Plasmodium* spp. carriage, the GAMM found that a positive PCR test was associated with age (p = 0.032) (Fig 6), travelling to the OIT (OR = 1.76, p = 0.023), and high vector density around a participant's home (high risk *versus* low risk OR = 4.11, p<0.001). Among outdoor activities, only farming was associated with a positive PCR test (OR = 1.96, p = 0.013) (Table 3).

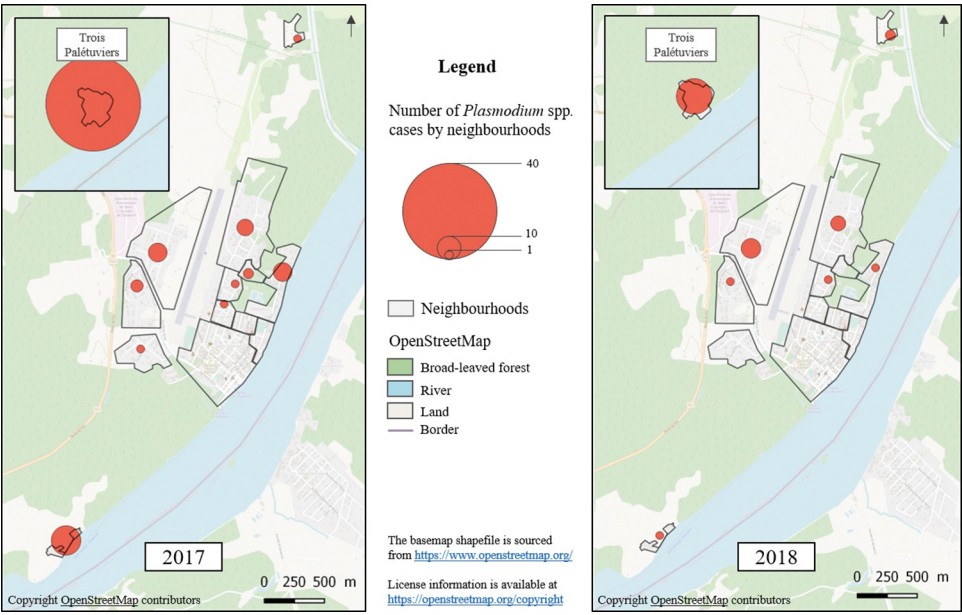

**Fig 5. *Plasmodium* spp. carriage prevalence in the study population in 2017 and 2018 according to neighbourhood.**

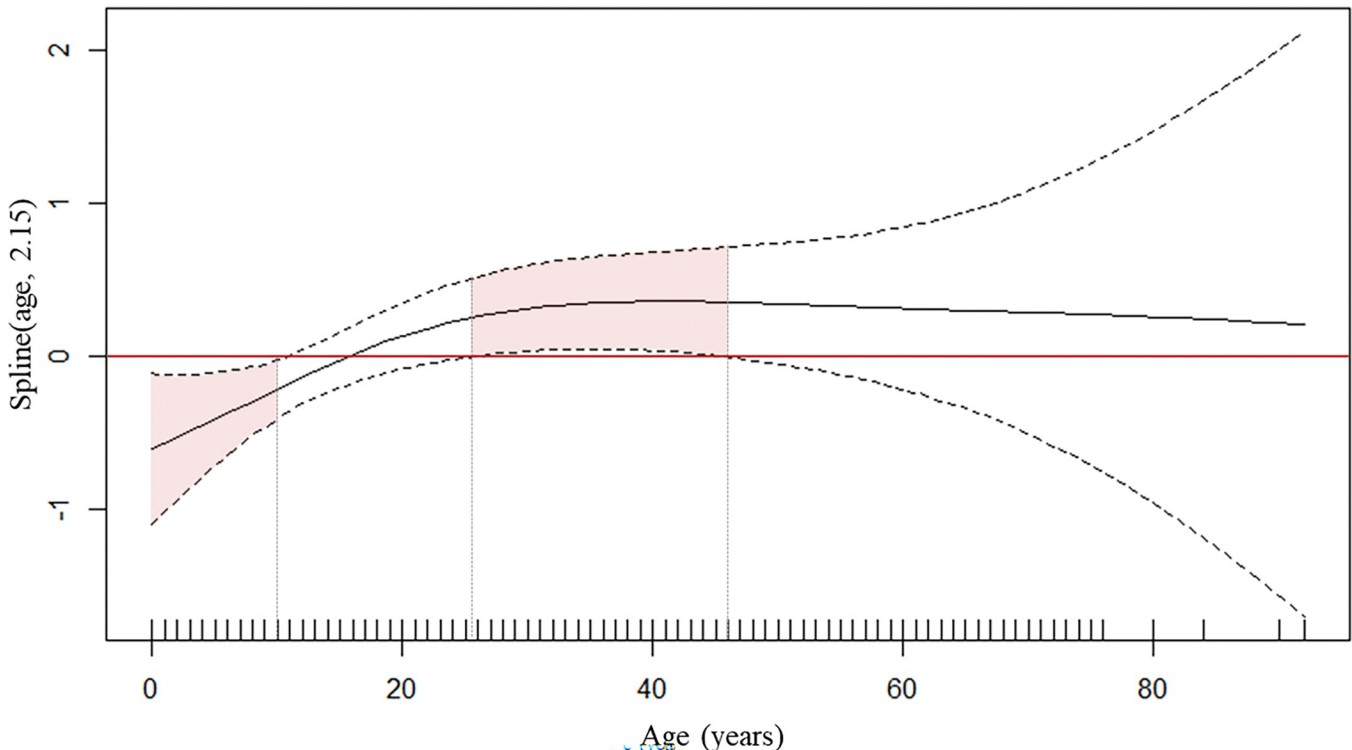

**Fig 6. Generalized additive model for relationship between *Plasmodium* spp. carriage and age.** The solid black line represents the *Plasmodium* spp. carriage/no *Plasmodium* spp. carriage ratio as a function of age. When positive, it means that the proportion of *Plasmodium* spp. in the study population was higher than the proportion of no *Plasmodium* spp.. The two dashed curved lines indicate the two standard error bounds.

## Discussion

### Key results

In this longitudinal ancillary study of persons living in SGO in French Guiana on the border with Brazil, we found that a higher estimated vector density around a participant's home was associated with a higher risk of *Plasmodium* spp. carriage, as were several types and levels of mobility, primarily occasional travel to a high-risk area—specifically the OIT—and regular peri-domestic mobility for farming.

**Table 3. GAMM model results for the impact of different variables on the risk of *Plasmodium* spp. Carriage.**

| Variable | Modality | OR | CI95 | P-value |
|---|---|---|---|---|
| **Male** | | 1.15 | 0.7–1.9 | 0.575 |
| **Amerindian ethnic group** | | 0.82 | 0.48–1.4 | 0.469 |
| **Vector density (ref: low)** | Medium | 0.92 | 0.37–2.29 | 0.855 |
| | High | 4.11 | 2–8.48 | < 0.001 |
| **Travel to Oiapoque Indigenous Territories** | | 1.76 | 1.08–2.88 | 0.023 |
| **Slash-and-burn farming** | | 1.96 | 1.15–3.34 | 0.013 |
| **Hunting** | | 0.97 | 0.5–1.88 | 0.93 |
| **Fishing** | | 1.00 | 0.59–1.71 | 0.986 |
| **Visiting a gold mining site** | | 1.47 | 0.62–3.53 | 0.382 |

## Interpretation

The associations found could be explained by the diversity of contamination paths when the vector is present in the living zone and in visited zones. These two different transmission chains have been observed both at a larger scale in all countries affected by malaria (mix of indigenous and imported cases), and a smaller scale, for example in Brazil [39]. Another study aiming to determine the micro-geographical heterogeneity of *P. vivax* parasitaemia in communities of the Peruvian Amazon found a clustered parasitaemia at both the household and community levels with complex transmission patterns related to human mobility among communities in the same micro-basin [40].

With regard to the association between *Plasmodium* spp. carriage and travel to the OIT, our study reinforces the suspicion that this particular cross-border mobility contributed to the malaria outbreaks in SGO in 2017 and 2018. A study on the 2017 outbreak found the peak *An. darlingi* density occurred at the same time in Trois Palétuviers and in the OIT. It also found a possible propagation for the spread from OIT to SGO based on the spatiotemporal distribution of cases [14]. In our study, 17% of the participants, mainly Amerindians, travelled to the OIT. This relatively high figure was to be expected, as most SGO Amerindian ethnic groups are originally from OIT and the cultural and spiritual connections are still strong there [28,41–43].

Our results on the association between *Plasmodium* spp. carriage and travel to the OIT reflect findings in studies worldwide highlighting the impact of episodic cross-border mobility (whether for work or to visit family) on malaria in low-transmission border areas [44–47].

With respect to mobility in the peri-domestic area, the only outdoor activity associated with *Plasmodium* spp. carriage in our study was slash-and-burn farming; this reflects previous findings in similar border areas in Brazil with Venezuela and Peru [48–50]. This result may be partly explained by collective work, as some people share slash-and-burn fields and help each other for specific work-related activities. As slash-and-burn farming is regular and occurs in the same place, there is an increased risk of being bitten by the same vectors. This outdoor activity, and the others we investigated (hunting and fishing), are linked to economic factors, and might increase for subsistence reasons when there is an economic crisis. Poverty is a characteristic of life at the French Guiana-Brazil border, with a high percentage of people working informally or practicing subsistence farming [51].

We found no direct association between *Plasmodium* spp. carriage and gold mining in this study; having said that, few participants reported visiting gold mining sites. This result was unexpected as the direct link between gold mining and malaria in French Guiana has already been highlighted, and the town of Oiapoque is a known supply and support base for goldmining activities [15,31,52]. There may however be an indirect role of gold mining in the malaria circulation in the Oyapock basin.

However, other factors were clearly associated with *Plasmodium* spp. carriage in our study, and these provide us with a much greater understanding of the persistence of malaria in French Guiana. Specifically, other types of cross-border mobility as well as estimated high vector density around a person's home also seem to have contributed to the 2017 and 2018 epidemics in SGO. These factors must be considered in future interventions to ensure they do not contribute to future epidemics or reintroduction.

## Public health implications

In contexts with low or very low malaria transmission, reactive case detection and treatment and reactive indoor residual spraying have been shown to be pertinent and effective, as has targeted mass drug administration [39,53–55]. Looking at our results, these three strategies, if implemented in SGO, should consider both the area around the targeted population's homes

and the places these people have visited when the vector is present (e.g., slash-and-burn fields, medium to high vector density areas). Other strategies could also be used, such as offering chemoprophylaxis (free of charge if possible) to persons travelling to high risk areas. This is a common practice in malaria-free countries [56]. Another option is to offer free PCR screening after returning home from a high-risk area. All these strategies are particularly important in eliminating malaria and preventing its reintroduction in SGO, given the large decrease in incidence in the area since the 2017 and 2018 epidemics, and the high rate of asymptomatic cases there [18,57].

Implementing these strategies effectively would necessitate facilitated access to malaria testing (e.g., reactive case detection, screening after travel in high-risk areas). This need could be met by community health workers (CHW) in the three remote neighbourhoods of SGO, a strategy already implemented in other areas worldwide. However, this would require a modification to French law, as malaria testing by non-medical health workers is not permitted anywhere in French territory. It would also require RDT with improved sensitivity [58].

Implementing these type of targeted reactive interventions would also require a reactive cross-border epidemiological surveillance system able to rapidly detect increased vector presence, identify any signs of a possible epidemic, and capture human population mobility and behavioural changes [59–61].

The path of the river Oyapock defines the French Guiana-Brazil border. There is a long history of cross-border mobility here, from daily or episodic to long term migration, all strongly linked to environmental, cultural and economic factors [34] (Fig 7). Given this context, the existing French-Brazilian binational partnership needs to be reinforced to create a reactive cross-border epidemiological surveillance system, and to ensure effective control strategies. More specifically, surveillance data need to be shared more comprehensively (currently, data are only shared on a monthly basis at the provincial level), and malaria control strategies harmonized (particularly in terms of the guidelines for dispensing drugs targeting *P. vivax* hypnozoites and access to testing) [5,7,19,45,48,62]. Furthermore, more detailed research is needed on how to implement malaria control strategies; cultural and economic specificities must be taken into account, as such strategies might not be perceived as a priority by the population or by healthcare workers [63]. Mobile CHW teams could become key actors in all steps of an intervention; their in-depth knowledge of the community could help to reinforce the reactivity of the surveillance system, lead to a better understanding of community perceptions, and adapt interventions to community-specific needs [52].

## Limitations

Our study has several limitations. The first concerns the analysis of mobility. As this was a longitudinal study with only two assessment points, it is hard to generalize the answers to the questions regarding mobility as constituting habits. Moreover, some data regarding trips in high risk areas were collected only in one survey. Discussions with inhabitants and healthcare workers in the area reinforced the hypothesis that SGO inhabitants usually go to the same places in the same time period, whether for work or to visit family.

Second, certain types of mobility were not investigated in our study; however, these were mainly related to places visited during daytime hours when the risk of bites from *An. darlingi* is very low (e.g., a day trip to the town of Oiapoque), and places with low *An. darlingi* density (e.g., a trip to the city of Cayenne). Moreover, no data was collected on the frequency of travel to high-risk areas.

Third, our study population cannot be regarded as representative of all SGO inhabitants, as only one third participated in the PALUSTOP study, and one quarter were included in our present analyses. The inhabitants of 3 of SGO's 15 neighbourhoods were not targeted by PALUSTOP's MTAT intervention, as malaria prevalence was lower there.

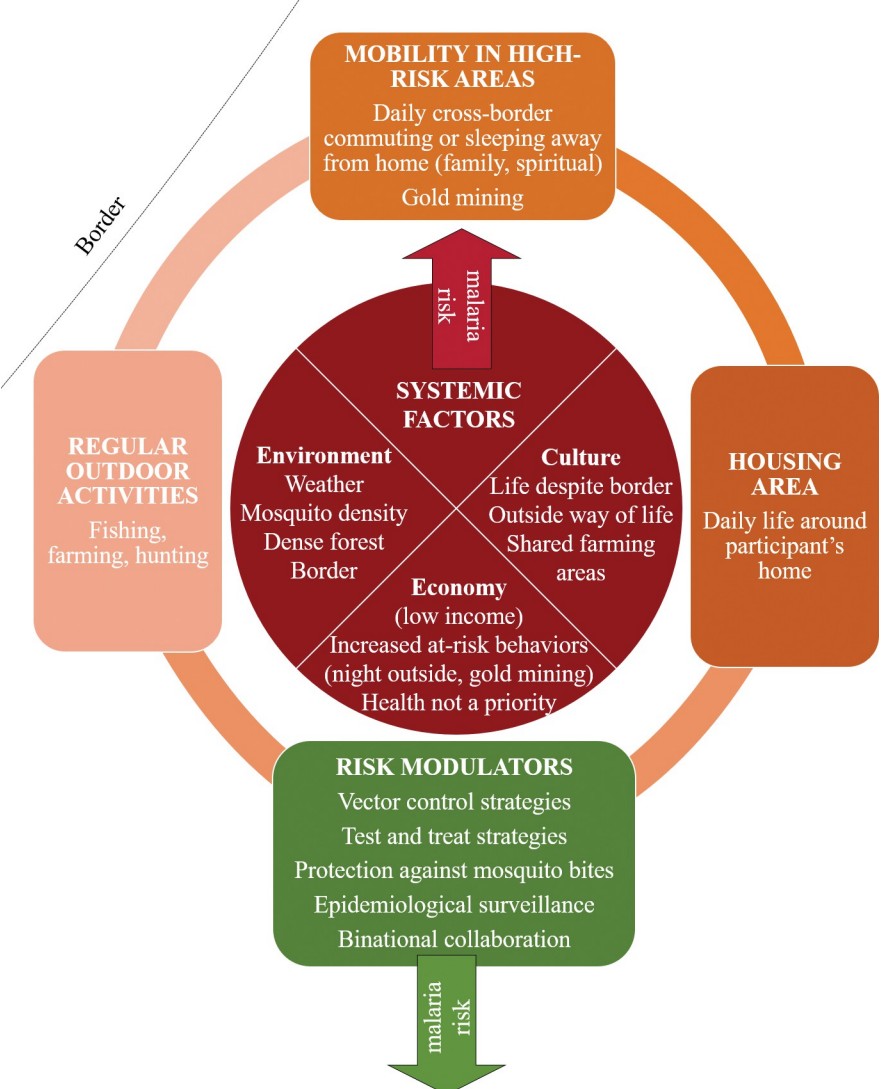

**Fig 7. Risk factors for *Plasmodium* spp. carriage in SGO.**

Finally, we analysed the data without considering the treatment received by *Plasmodium* spp. carriers in 2017 within the framework of the PALUSTOP's MTAT campaigns. This probably induced selection bias, as a percentage of the tests analysed (i.e., some of the 2018 tests) came from participants treated in 2017 as part of the MTAT campaign who might otherwise not have been treated (i.e., asymptomatic). However, we assume that this bias was sufficiently low that the analysis of both years together could be performed with enough statistical power being guaranteed. Only positive PCR were treated in 2017, meaning that only some of the people carrying hepatic *P. vivax* hypnozoites were detected and treated.

## Conclusion

This study highlights the complexity of the factors associated with *Plasmodium* spp. carriage in the border area between French Guiana and Brazil, an area with low transmission. The persistence in cases may be associated with different risk factors which cumulatively contribute to

incidence. These include peri-domestic mobility and mobility to high-risk areas, as well as high vector density around persons' homes. These specificities need to be considered when designing specific surveillance systems and targeted interventions with a view to disease elimination. Cross-border cooperation and community participation seem to be particularly appropriate strategies to use in this context, especially as they help to create a greater understanding of the changing context and community perceptions. They can also reinforce the surveillance system in the complex setting of border areas.

## Supporting information

**S1 Table. Sociodemographic breakdown and risk factors for the study population by neighbourhood in Saint Georges de l'Oyapock.**
(DOCX)

**S2 Table. Sociodemographic characteristics and exposure according to PCR result in 2017 and 2018.**
(DOCX)

## Acknowledgments

The authors are very grateful to the study participants in SGO, French Guiana, to Olivier Moriceau and the community workers of the DAAC (Non-Governmental Organization) who provided crucial support throughout this study, and to Dr. Franck de Laval, Dr. Gaëlle Walter, Dr. Bastien Bidaud, Dr. Alessia Melzani, Dr. Céline Michaud and Mrs. Mylene Cebe who all participated in data collection. Finally, our thanks to Mrs. Charlène Cochet and Mrs. Roziane Silva Barbosa who participated in data contextualization and to Jude Sweeney (Milan, Italy) for the English revision and copyediting of the manuscript. Map data use OpenStreetMap® contributors. OpenStreetMap® is *open data*, licensed under the Open Data Commons Open Database License (ODbL), available at https://www.openstreetmap.org/copyright.

## Author Contributions

**Conceptualization:** Lise Musset, Felix Djossou, Emilie Mosnier.

**Formal analysis:** Hélène Tréhard, Yassamine Lazrek, Jean Gaudart.

**Funding acquisition:** Lise Musset, Emilie Mosnier.

**Investigation:** Lise Musset, Loïc Epelboin, Emilie Mosnier.

**Methodology:** Lise Musset, Emilie Mosnier.

**Project administration:** Lise Musset, Emilie Mosnier.

**Supervision:** Lise Musset, Felix Djossou, Jean Gaudart, Emilie Mosnier.

**Validation:** Lise Musset, Emilie Mosnier.

**Writing – original draft:** Hélène Tréhard.

**Writing – review & editing:** Hélène Tréhard, Lise Musset, Yassamine Lazrek, Felix Djossou, Loïc Epelboin, Emmanuel Roux, Jordi Landier, Jean Gaudart, Emilie Mosnier.

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
