## [Decision Letter · Decision Letter 0]

10 Oct 2023

PGPH-D-23-01522

The impact of mobility on Plasmodium spp. carriage in border areas with a low malaria transmission rate: an example in Amazonia

Dear Dr. TRÉHARD,

Thank you for submitting your manuscript to PLOS Global Public Health. After careful consideration, we feel that it has merit but does not fully meet PLOS Global Public Health’s publication criteria as it currently stands. Therefore, we invite you to submit a revised version of the manuscript that addresses the points raised during the review process.

We look forward to receiving your revised manuscript.

Kind regards,

André Machado Siqueira, M.D., MSc, Ph.D

Academic Editor

Journal Requirements:

1. Some material included in your submission may be copyrighted. According to PLOS’s copyright policy, authors who use figures or other material (e.g., graphics, clipart, maps) from another author or copyright holder must demonstrate or obtain permission to publish this material under the Creative Commons Attribution 4.0 International (CC BY 4.0) License used by PLOS journals. Please closely review the details of PLOS’s copyright requirements here: PLOS Licenses and Copyright. If you need to request permissions from a copyright holder, you may use PLOS's Copyright Content Permission form.

Potential Copyright Issues:

"Figure 1-3: please (a) provide a direct link to the base layer of the map (i.e., the country or region border shape) and ensure this is also included in the figure legend; and (b) provide a link to the terms of use / license information for the base layer image or shapefile. We cannot publish proprietary or copyrighted maps (e.g. Google Maps, Mapquest) and the terms of use for your map base layer must be compatible with our CC-BY 4.0 license. 

"

Additional Editor Comments (if provided):

Dear Dr Tréhard,

We appreciate the submission of your manuscript which both reviewers agree have importance in the malaria field. However there are a few issues that need to be revised, mainly pointed by reviewer #2 which that should receive your attention. We expect a new version of the manuscript.

Best regards

Reviewers' comments:

Reviewer's Responses to Questions

**Comments to the Author**

1. Does this manuscript meet PLOS Global Public Health’s publication criteria? Is the manuscript technically sound, and do the data support the conclusions? The manuscript must describe methodologically and ethically rigorous research with conclusions that are appropriately drawn based on the data presented.

Reviewer #1: Yes

Reviewer #2: Yes

2. Has the statistical analysis been performed appropriately and rigorously?

Reviewer #1: Yes

Reviewer #2: Yes

3. Have the authors made all data underlying the findings in their manuscript fully available (please refer to the Data Availability Statement at the start of the manuscript PDF file)?

Reviewer #1: No

Reviewer #2: No

4. Is the manuscript presented in an intelligible fashion and written in standard English?

Reviewer #1: Yes

Reviewer #2: Yes

5. Review Comments to the Author

Reviewer #1: A well planned, structured study with a clear and key research question or problem: Cross-border malaria and the elimination route in the region. The discussion of the results is pertinent as well as the recommendations derived from the results.

Reviewer #2: The authors have presented a study holds significant importance in the context of malaria elimination in cross-border areas. It highlights the challenges faced in achieving and maintaining malaria elimination, particularly in regions like Amazonia. The study reveals that despite a substantial reduction in malaria incidence over the past decade, the final steps toward elimination remain the most difficult, especially in international border regions. Key findings of the study underscore the crucial role of mobility in Plasmodium spp. carriage. It identifies factors such as travel to adjacent Indigenous territories, vector density around participants’ homes, slash-and-burn farming, and age as contributing to malaria transmission tailored to different types of mobility. Additionally, the study’s implications are significant for public health strategies, suggesting that strategies like reactive case detection and treatment in areas frequently visited by mobile populations are essential in cross-border regions. By recognizing the impact of mobility on malaria transmission and tailoring interventions accordingly, this reach contributes to the ongoing efforts to eliminate malaria, particularly in challenging border areas.

However, the study has some limitations, such as data from only two years and entomology data was estimative. Thus, a principal suggestion is changing the title for “Understanding the Impact of Mobility on Malaria Transmission in an Amazon Cross-Border Area with Low Transmission Rates”, because the real “impact” was not found and describe due data limitation, if you consider that you found the impact, describe in the abstract and in the conclusion section. Additionally, there are other points described below that need improve:

If you choose to retain the title, please note that “spp.” should not be italicized. The same observation applies to short title.

Abstract:

Line 38: spp. should not be italicized. Please, review the entire manuscript for this formatting.

Line 44: Why were patients treated only if it was possible?

Introduction

Line 66: P. falciparum is first time mentioned in the text, so you have to describe it without using the abbreviation.

Materials and methods

Line 114: Materials AND methods

Fig. 1: the quality of the figure is not good.

Fig 2: The legend of the figure is difficult to read and does not clearly identify each color.

Line 189: P. vivax at 0.5 parasite/ μL?

Line 205: Legend for Fig 3: check the scientific name of Anopheles darlingi.

Results

Fig 4. Check the font type used in some of text box.

Lines 252 to 257 and Fig. 4: Why you had only 1,192 participants who took part in both surveys have available PCR results for both 2017 and 2018 when you had 1,501 with PCR result in 2017 and 1,232 with PCR results in 2018, and you had only 290 lost follow-ups?

Line 261 to 263 and Table 01: is this information only form 2017? If not, why did you only describe the median age from 2017?

Table 1: why did you not include data for female in the Table?

Line 272: I suggest writing (Table 1 and S1 Table) instead of (S1 Table) (Table 1).

Table 2: I suggest providing information about Regina in the setting section, as you did for Oiapoque.

Lines 296 to 300: Is this information about 2017 and 2018? If so, why did you describe the percentage of travel to high-risk areas only for 2018?

Fig 5. The font used in the Legend is not easy to read, the quality of the figure is not good, and the circle indicating the number of cases of Plasmodium is unclear.

Line 324: Legend for Figure 6: Plasmodium should be written with a lowercase “p”.

Discussion

Line 342: Abbreviate the genus Plasmodium for the specie P. vivax.

Line 348 and 428: Abbreviate the genus Anopheles darlingi for the specie An. darlingi.

Fig 7. The quality is not good, and the text is not visible.

Line 387: This is a common practice in malaria-free countries. – Add the reference.

6. PLOS authors have the option to publish the peer review history of their article (what does this mean?). If published, this will include your full peer review and any attached files.

**Do you want your identity to be public for this peer review?** For information about this choice, including consent withdrawal, please see our Privacy Policy.

Reviewer #1: **Yes: **Maria Eugenia Grillet

Reviewer #2: No

---

## [Decision Letter · Decision Letter 1]

3 Jan 2024

Understanding the impact of mobility on Plasmodium spp. carriage in an Amazon cross-border area with low transmission rate

PGPH-D-23-01522R1

Dear Dr Gaudart,

We are pleased to inform you that your manuscript 'Understanding the impact of mobility on Plasmodium spp. carriage in an Amazon cross-border area with low transmission rate' has been provisionally accepted for publication in PLOS Global Public Health.

Best regards,

André Machado Siqueira, M.D., MSc, Ph.D

Academic Editor

Reviewer Comments (if any, and for reference):

Reviewer's Responses to Questions

**Comments to the Author**

1. If the authors have adequately addressed your comments raised in a previous round of review and you feel that this manuscript is now acceptable for publication, you may indicate that here to bypass the “Comments to the Author” section, enter your conflict of interest statement in the “Confidential to Editor” section, and submit your "Accept" recommendation.

Reviewer #2: All comments have been addressed

2. Does this manuscript meet PLOS Global Public Health’s publication criteria? Is the manuscript technically sound, and do the data support the conclusions? The manuscript must describe methodologically and ethically rigorous research with conclusions that are appropriately drawn based on the data presented.

Reviewer #2: Yes

3. Has the statistical analysis been performed appropriately and rigorously?

Reviewer #2: Yes

4. Have the authors made all data underlying the findings in their manuscript fully available (please refer to the Data Availability Statement at the start of the manuscript PDF file)?

Reviewer #2: Yes

5. Is the manuscript presented in an intelligible fashion and written in standard English?

Reviewer #2: Yes

6. Review Comments to the Author

Reviewer #2: (No Response)

7. PLOS authors have the option to publish the peer review history of their article (what does this mean?). If published, this will include your full peer review and any attached files.

**Do you want your identity to be public for this peer review?** For information about this choice, including consent withdrawal, please see our Privacy Policy.

Reviewer #2: No
